# Recent Trends in Non-Invasive Methods of Diagnosis and Evaluation of Inflammatory Bowel Disease: A Short Review

**DOI:** 10.3390/ijms25042077

**Published:** 2024-02-08

**Authors:** Dan Vălean, Roxana Zaharie, Roman Țaulean, Lia Usatiuc, Florin Zaharie

**Affiliations:** 1Regional Institute of Gastroenterology and Hepatology “Octavian Fodor”, 400162 Cluj-Napoca, Romania; valean.dan@elearn.umfcluj.ro (D.V.); axiromplus@gmail.com (R.Ț.); florinzaharie@yahoo.com (F.Z.); 2Department of General Surgery, University of Medicine and Pharmacy “Iuliu Hațieganu”, 400347 Cluj-Napoca, Romania; 3Department of Gastroenterology, University of Medicine and Pharmacy “Iuliu Hațieganu”, 400347 Cluj-Napoca, Romania; 4Department of Patophysiology, University of Medicine and Pharmacy “Iuliu Hațieganu”, 400347 Cluj-Napoca, Romania; lia.usatiuc@yahoo.com

**Keywords:** inflammatory bowel disease, diagnosis, prognosis, non-invasive

## Abstract

Inflammatory bowel diseases are a conglomerate of disorders causing inflammation of the gastrointestinal tract, which have gained a significant increase in prevalence in the 21st century. As they present a challenge in the terms of diagnosis as well as treatment, IBDs can present an overwhelming impact on the individual and can take a toll on healthcare costs. Thus, a quick and precise diagnosis is required in order to prevent the high number of complications that can arise from a late diagnosis as well as a misdiagnosis. Although endoscopy remains the primary method of evaluation for IBD, recent trends have highlighted various non-invasive methods of diagnosis as well as reevaluating previous ones. This review focused on the current non-invasive methods in the diagnosis of IBD, exploring their possible implementation in the near future, with the goal of achieving earlier, feasible, and cheap methods of diagnosis as well as prognosis in IBD.

## 1. Introduction

Inflammatory bowel diseases (IBD) are pluri-factorial entities, consisting mostly of Ulcerative Colitis (UC) and Crohn’s Disease (CD) which can manifest through chronic inflammation of various regions of the digestive tract. Although the mechanism is primarily immune, through various alterations of the immune response against different triggers, there are also genetic, infectious (and microbiota-related), and environmental factors that need to be considered [1]. The increasing incidence in developed and developing countries, especially in Europe and North America, as well as a tendency for diagnostic delays in some of the regions can prompt the need for faster, cheaper, non-invasive methods of diagnosing as well as evaluating inflammatory bowel diseases [2]. There has been a marked increase in IBD cases, especially in the pediatric sector; 25% of the diagnosed population present symptoms before 20 years of age. Early diagnosis of IBD has been linked with an improved prognosis, as well as showcasing better long-term survival [2,3].

There is a fine line between diagnosing a patient with either irritable bowel syndrome (IBS) or an IBD. Although they might show similarities, regarding the microbiota, some of the genetic factors, as well as psychological factors, there are major differences regarding the severity of the inflammation as well as the presence of certain biomarkers such as fecal calprotectin [3]. Recently, non-invasive methods for both diagnosis and prognosis of IBD have been gaining popularity. From scoring methods to serological tests, spectroscopy as well as various fecal markers, overlapped with invasive methods such as colonoscopy, and various techniques of imaging, the differentiation of the previously mentioned entities is much easier to performed [4,5,6]. Despite this, IBD remains a challenging pathology with a relatively high economic impact on current healthcare, thus prompting the need for further research regarding cost-effective and reliable methods of detection [5].

The differentiation between the entities continues to remain an issue in contrast with the presence of other autoimmune diseases as well as various dysbiosis and gastrointestinal infections, especially in patients with an early onset of the disease, thus further prompting the need for a non-invasive diagnostic tool. The purpose of this systematic review was to review the current literature regarding the available non-invasive tools in terms of diagnosis and follow-up of IBDs.

## 2. Material and Methods

A systematic search of three databases (PubMed, Scopus, and Embase) was performed, with no language restrictions, highlighting the studies on the topic of “diagnosis and prognostic of IBDs”. We selected all the studies that were published between January 2003 and December 2023. The key words non-invasive, ulcerative colitis, Crohn’s Disease, inflammatory bowel disease, serology, prognostic, diagnostic methods, as well as their equivalent terminology were used.

## 3. Results and Discussion

### 3.1. Scoring Systems in Inflammatory Bowel Disease

Although most of the early scoring systems for both UC and CD were developed in the 20th century [7,8], they can still serve as a means of evaluating the severity in both pathologies. The most used scoring systems in the current literature are the Crohn Disease Activity Index (CDAI) used for CD and the Mayo Score (Disease Activity Index) used for UC. Although they present with a moderate degree of specificity, they are widely used in establishing the severity of the disease flare.

The Crohn’s Disease Activity Index (Table 1), developed in 1976 by Best et al. [7], remains the “gold standard” for the evaluation of disease activity. Although maintaining a low specificity, it retains its usefulness in clinical trials, when evaluating post-treatment disease activity. It is comprised of eight variables, each identified as reliable predictors: average number of stools, usage of antidiarrheal agents over the last 7 days, average abdominal pain rating, general well-being over 7 days, presence of extraintestinal complications, palpation of abdominal mass, deviation in hematocrit, and deviation in body weight. Based on an algorithm, the CDAI score can present values from 0 to 600, with a score less than 150 corresponding to relative remission of the disease, 150–219 mild disease, 220–450 moderate disease, and over 450 indicating severe activity of the disease. Moreover, a variant of CDAI was developed by Harvey and Bradshaw [9] which uses only five variables without the requirement of keeping track of 7-day variables as well as laboratory values, while maintaining a high degree of correlation with the CDAI score (r = 0.93). Moreover, a study by Vermiere et al. [10] was able to support the correlation between the two scores while highlighting the simplicity of the latter, thereby being equally effective in the assessment of disease severity.

Despite their applicability in a clinical trial setting as well as in a clinical setting, with mobile applications being developed for patients to self-administer the tests to further reduce sanitary costs [11], as well as a high degree of accessibility for both physicians and patients alike, with the development of patient-based indexes in contrast with clinician-based indexes [12], studies have highlighted the limitations of these scores. One of the primary issues stems for the subjectivity of three out of seven criteria (keeping track of the number of stools, abdominal pain, as well as the sense of well-being), thus some of the measurements are subjected to bias. In addition, Peyrin-Biroulet et al. [13] highlighted a discrepancy between endoscopic healing and CDAI. In a clinical trial involving 508 patients that compared infliximab to azathioprine, clinical symptoms measured using the CDAI were not considered a reliable measure of the underlying inflammation in CD. Thus, the subjectivity of clinical symptoms should not influence the decision regarding medical therapy. Another limitation of clinical scores is the lack of any inflammatory markers involved in the grading of the pathology. As a response to the subjectivity of the score, further studies evaluated the presence of a modified Crohn’s disease activity index in which subjective measurements were removed, especially in CD patients with enterostomy. Thus, Ishida et al. developed a modified Crohn disease activity index (mCDAI), removing the defecation frequency. It had a positive correlation with the inflammatory markers, fully reflecting the nutritional status and inflammatory response [14].

Since the scoring proposed by Truelove and Witts (Table 2) [15] in ulcerative colitis, which was comprised of six variables (number of stools, fever, tachycardia, hemoglobin level, ESR, and weight loss/gain), several severity scores have been developed to increase their clinical correlation and to further describe the severity of the disease. Peyrin-Biroulet et al. described three main domains which are relevant to the evaluation of the severity of the disease: impact of the disease on the patient (comprising clinical symptoms, patient-reported outcomes, quality of life, and disability), inflammatory burden (location, extent, and severity of bowel involvement) and disease course, including structural damage [16]. One of the most popular scores used in clinical practice regarding the evaluation of ulcerative colitis is the Mayo Score. It comprises the stool frequency, rectal bleeding, findings of flexible sigmoidoscopy, as well as the global assessment of the physician, ranging from 0–12 (three points for each unit) [17].

One of the major disadvantages of the score was the minimally invasive component which would hinder the capability of applying it in a clinical trial setting. Lewis et al. expanded on this problem, by proposing a non-invasive partial Mayo Score, which can be used as an outcome measure for clinical trials [18]. The partial Mayo Score was applied to 105 patients enrolled in a 12-week, randomized, placebo-controlled trial. Each index was strongly correlated with the patient’s rating of the disease; the partial tests obtained a sensibility and specificity of over 80 percent. Thus, the partial Mayo Score comprised only of the stool frequency and bleeding can indicate the patient-perceived clinical response as well as the complete Mayo Score. In addition, Sandborn et al. proposed a modified Mayo Score (mMayo) which excludes the subjectivity of the physical global assessment, by applying it over two 8-week induction studies (OCTAVE induction 1 and 2) and a 52-week maintenance study (OCTAVE sustain) [19]. Thus, a significant effect of tofacitinib versus placebo was highlighted using the mMayo, which is consistent with the previously reported data using the original Mayo Score.

In conjecture with the clinical trial scores, endoscopic scores have been developed for both pathologies, namely the Crohn Disease Endoscopic Index of Severity (Table 3) (CDEIS), proposed by Mary et al. in 1989 [20], as well as the Ulcerative Colitis Endoscopic Index of Severity (UCEIS), more recently proposed by Travis et al. in 2012 [21]. Despite being straightforward and dependent on a minimally invasive procedure, these scores can better assess the clinical evolution of the patient by highlighting mucosal healing, which is viewed as one of the important treatment goals in IBD as well as a predictor of long-term outcomes [22,23,24]. Colonoscopy has an important role in diagnosing as well as monitoring IBD [25]; however, the main issue that investigators have been confronted with over the years is maintaining a low intra and inter-investigator variability. This was, furthermore, reflected in a second study by Travis et al. [26]. In addition, Ikeya et al. highlighted a higher accuracy of predicting the clinical outcomes as well as long-term prognosis using the UCEIS in comparison with the Mayo Score over a cohort of 41 patients. Although there are other variants of endoscopy scores for both pathologies, they can present a low validation rate as well as high variations, minimizing their applicability in an inter-investigator context. Studies suggest a simplification of endoscopic indices in terms of variability and reliability, as well a general assessment of the histological status [27].

### 3.2. Serological Markers

Serological markers have been on the rise in the past 15 years, due to their increased accessibility in a clinical setting. The presence of such markers might be a consequence of the dysregulated immune response to various gut microbiota [28]. In addition, the increased need to clinically differentiate UC from CD can prove difficult as some of the clinical symptoms are difficult to distinguish, as well as a part of endoscopic features may not be clear. Primarily, antibodies related to IBD can comprise two categories: antibodies targeting microbial antigens (most known are anti-glycan antibodies, anti-OmpC anti-I2 and Anti-Cbr1), and autoantibodies (consisting most commonly of pANCA, PAB, and GAB) [29].

The most well-known and studied anti-glycan antibodies are, to this date, the anti-neutrophil cytoplasmic antibodies (ANCAs) and anti-Saccharomyces cerevisiae antibodies (ASCA). Although ANCA is traditionally associated with vasculitis as well as various rheumatological pathologies, the first usage of ANCA in IBD patients was reported in a cohort with ulcerative colitis dating back to 1990 [30]. Primarily, the prevalence of ANCA is higher in patients with UC than CD, as highlighted by various cohort studies, with differences of prevalence based on geographic regions [31,32,33], reporting a detection of up to 70% in UC cases compared to 10% in CD [34].

ASCAs are antibodies for the mannan hydro-polimer found in the cell wall of Saccharomyces cerevisiae, which occurs mostly in patients with CD (up to 60–70%) but very rarely in UC, having a prevalence of only 4% [35]. The first usage of ASCA in IBD was highlighted in a study by Saxon et al., back in 1990 [36]. ASCA has been confirmed through various studies to have a low sensitivity, yet high specificity (up to 96%) in CD, with both IgG and IgA antibodies being present in the serum of the selected patients [33,34].

When confronted with differentiating UC from CD and from indeterminate colitis (IC), a combination of antibodies can be more effective in establishing the final diagnosis. Bossiut et al. highlighted that a combination of ASCA and ANCA can effectively differentiate CD from UC, due to the different patterns of the antibodies’ combination. Thus, patients with CD were positive for ASCA and negative for p-ANCA; on the other hand, patients with UC were positive for p-ANCA and negative for ASCA [37]. Moreover, Mokrowiecka et al. showcased similar results over a cohort of 125 patients with UC, CD, or IC and 45 patients with functional intestinal disorders [38]. Thus, the prevalence of pANCA was higher in the UC group (68%), and ASCA was significantly higher in the CD group (81%). Moreover, the combined panels of the markers highlighted a specificity of up to 100% in the pANCA+/ASCA− group in patients with UC and 92% for patients with pANCA−/ASCA+ group in patients with CD, respectively. The conclusion of these studies was that serological markers are much more effective in differentiating the pathologies rather than diagnosing the diseases. In addition, Zhou et al. highlighted that the most important utility of the serological markers in IBD is to predict the course of the disease as well as highlight the aggressive phenotypes, thus pointing out the prognostic role of the inflammatory markers over their diagnostic role [39].

Recently, in addition to ASCAs, more anti-glycan antibodies have shown their effectiveness as diagnostic biomarkers in IBDs. The most noteworthy ones are anti-chitobioside carbohydrate IgA antibodies (ACCAs), antilaminarbioside carbohydrate IgG antibodies (ALCAs), and anti mannobioside carbohydrate IgG antibodies (AMCAs), highlighted by Dotan et al. in 2006. [40] by profiling antibodies from the serum of patients diagnosed with IBD using glycan array technology and ELISA. The basic elements from which these antibodies are derived are components of the cell walls of yeast, fungi, and bacteria prevalent in our gut microbiota, capable of driving an innate immunity response. Moreover, Dotan et al. highlighted the capability of discriminating CD from UC using these biomarkers with a specificity of over 90% [40], with a specificity of up to 99% when combining any two of these antibodies. In addition, higher levels of ALCA and AMCA are associated with small intestine lesions [40,41]. Furthermore, two other novel anti-glycan antibodies have been discovered, namely anti-laminarin IgA antibody (Anti-L) and anti-chitin IgA antibody (anti-C) [42]. Although they present a low degree of sensitivity, their specificity is relatively high (studies reporting over 90%) [43]. In addition, anti-L and Anti-C are strongly associated with the need for IBD related surgery, thus being a useful prognostic marker [44]. Recent studies have shown that combinations of these antibodies and ASCA may not be useful tools in the classification of IBD [45]. A study by Paul et al., however, underlined that higher values of ACCA and anti-laminarin antibodies are frequently associated with the need to use steroids in IBD patients. They have also defined the severity of CD based on elevated AMCA, ASCA, and ACCA, whereas a severe case of UC is determined via elevated levels of AMCA and ACCA [46]. In addition, ACCA may be associated with the penetrating and structuring behavior of the disease. [47].

Out of the antimicrobial antigens discovered, the most widely used in the differentiation of CD from UC is the antibody to the outer membrane porin C of Escherichia Coli (anti-OmpC), originally identified as a cross-reactive antigen of pANCA [48]. Studies revealed a low sensitivity and high specificity (20–50% vs. 88%) in CD [49,50]. One of the advantages of anti-OmpC is diagnosing CD patients with negative ASCA, as the percentage of OmpC+/ASCA− is around 15% [51]. In addition, anti-OmpC can be considered a marker of severity in CD, as an association with ANCA and anti-I2 is more likely to develop penetrating diseases [52]. Anti-I2 is another novel antibody against bacterial antigens, namely Pseudomonas fluorescens which is generated from PfiT, a gene accounting for bacterial T-cell super antigenic activity. It has also been positively linked to CD, with anti-I2 being significantly positive in over 50% cases of CD compared to 10% of cases in UC, thus being an effective marker for differentiating IBD [34]. In a similar fashion, flagellin Cbir1 presents a colitis inducing role, which has been shown in experimental studies on mice. It has been cloned as a colitis-associated bacterial antigen [53], being positive in over 50% of CD cases, compared to 10% of UC cases [53,54,55].

When accounting for more than one antibody, OmpC antibodies have shown the highest specificity for needing surgery, and ASCA presents the highest sensitivity for the surgery, as shown in a meta-analysis by Xiong et al. in 2014 [56]. Compared to the rest of the antimicrobial agents, anti-Cbir1 is significantly associated with structuring and penetrating disease, in association with anti-I2 [57]. Both markers are also related to disease duration and early postoperative recurrence [58,59]. Anti-CBir1 is also associated with severe phenotypes in pediatric CD [60,61]. In addition, Hamilton et al. highlighted in a prospective study that the patients who underwent two or more surgeries were more likely to be OmpC positive [62]. The main issue, however, remains the low sensitivity of these antibodies in the diagnosis and prognosis of IBD, thus their role remains controversial.

Another group involved in the prognostic role of IBD in terms of serological markers is the antibodies against the exocrine pancreas (PABs). Although they were first described by Stöcker et al. in 1987 [63], who demonstrated that PABs were found mainly in CD patients, recent studies have highlighted the importance of glycoprotein 2 (GP2) and CUB and zona-pellucida-like domain 1 (CUZD1) [64,65].

GP2 is a membrane-bound receptor, located mostly in the intestinal Peyer’s patches, involved in the mediation of the bacteria-specific mucosal response. It is rarely found in the colon; therefore, the release of GP2 is mostly related to small bowel inflammation, thus accounting for a higher expression in CD [66]. Anti-GP2 consists of anti-GP2 IgG and anti-GP2 IgA, which can eliminate the immune responses of GP2 and can improve the transcytosis effect by introducing a GP2-bacteria complex into M-cells, as highlighted by Ohno et al. [67]. In addition, CD patients with exclusive colonic involvement presented lower levels of anti-GP2 than patients with ileum involvement [68]. The biological role of CUZD1 remains unknown; some studies suggesting that anti-CUZD1 positivity is not correlated with the biological outcome [69]. Both antibodies are elevated in CD patients compared to UC patients. However, PAB usage in the diagnosis of IBD should be performed cautiously and in a clinical setting, as they may be detected in other autoimmune pathologies such as primary sclerosing cholangitis and cholangiocarcinoma [70].

Granulocyte macrophage colony-stimulating factor (GM-CSF) is a cytokine produced by lamina–propria immune cells, playing a part in the process of maturation in antimicrobial functions of myeloid cells [71]. Studies have associated the anti-GM-CSF antibodies with an aggressive pattern and ileal development in patients with CD [72], thus being useful in differentiating CD from UC. In addition, elevated anti GM-CSF antibodies may indicate a more rapid rate of progression towards complications especially in patients with small bowel involvement [72].

Recent developments have highlighted the role of MicroRNAs (MiRNAs) in the activity and diagnosis of IBD. MicroRNAs are non-coding RNAs which can regulate gene expression post-transcription [73]. Their potential role as non-invasive biomarkers is given by their stability in circulation due to their short length. In a recent study published by Jung et al., the role of Mi-RNA is adequately described and divided into groups, being involved in the intestinal barrier in IBD as well as the immune response and autophagy [74]. Wang et al. highlighted the prognostic role of miR-223, which is increased in the inflamed colonic mucosa in patients with both CD and UC, and it correlates directly with the disease activity [75]. Zahm et al. highlighted 11 different Mi-RNAs (mi-R16, miR-484, miR-30e, miR-106a, miR-195, miR-20a, miR-21, miR-140, let-7b, miR-192, and miR-93) which may be useful in the diagnosis of IBD in children, with good predictability (AUC: 0.821–0.917; sensitivity 69.5–82.61%) [76]. The role of mi-RNA in the development of structuring CD has been highlighted by Nijhuis et al. (MiR-29a) and Lewis et al. (MiR-19a-3p and miR-19b-3p) [77,78]. While Paraskevi et al. highlighted six different miRNAs involved in the diagnosis of UC as well as eleven different miRNAs involved in the diagnosis of CD [79], Wu et al. showcased five miRNAs with elevated values in inflammatory bowel diseases [80]. One noteworthy aspect is the presence of MiR-199a-5p, which based on its level can be used as a diagnostic tool for IBD as well as a differentiation tool between UC and CD [79,80]. A recent review by Oliviera et al. highlights the role of miRNA in the modulation of microbiota in IBD [81].

### 3.3. Blood Inflammatory Markers

As both CD and UC may include flares of inflammatory disease, it is important to recognize these episodes as promptly as possible and to differentiate an IBD from either a functional disorder or irritable bowel syndrome. Despite not being known for their high specificity, the erythrocyte sedimentation rate (ESR) and C-reactive protein (CRP) may highlight disease activity in IBD.

Serum CRP is an acute-phase reactant, considered a good predictor and accepted as an inflammatory marker in IBD, despite its low specificity, as its value can be increased in various inflammatory conditions [82]. It has a short half-life between 16–24 h; therefore, it can adequately assess inflammatory activity in the organism. Due to its accessibility as well as cost-effectiveness, it is the most widely used marker for assessing IBD activity through repeated measurements [83]. Elevated CRP levels associated with symptoms in patients with CD can be associated with an elevated risk of recurrence [84]. It has, however, some drawbacks, as it is not disease specific and sometimes it can be elevated in non-IBD enteritis [85]. In addition, it presents a lower accuracy in patients with low activity of the disease, especially in CRP-negative patients [83]. Despite previous studies showing a good correlation in patients with CD, it cannot distinguish UC from CD. The sensitivity of CRP decreases in pediatric patients; therefore, it cannot be used as a prognostic marker, being unable to establish the levels of the activity [43]; some studies highlight normal values of CRP in active disease in over 25% of pediatric cases of CD [86]. One of the biggest drawbacks, however, stems from the fact that CRP cannot adequately predict mucosal healing. In a study published by Krzystek-Korpacka et al., large variations in sensitivity and specificity in both UC and CD in the detection of mucosal healing were highlighted [87].

When accounting for the prediction of the recurrence, several reports showcase CRP’s capability to predict clinical recurrence in CD. In a prospective cohort study performed by Roblin et al. in 2015, it was reported that elevated CRP levels (higher than 5 mg/L) 22 weeks after the introduction of therapy with infliximab accurately predicted the loss of response [88]. Several studies do not support this theory; however, Boschetti et al. highlighted a weak difference between patients with endoscopic recurrence and endoscopic remission over a median of 7 months [89].

When predicting the therapeutical effect, there are studies that support the usage of CRP as a marker of prognosis. Iwasa et al. highlighted, in a study of 72 patients, that the improvement in clinical symptoms with a reduction in CRP in the next two weeks was associated with an improved prognosis [90]. Similar results were obtained in the ACCENT trial, showcasing a decrease of 60% in CRP in 14 weeks, strongly associated with remission [91].

ESR is widely available and cheap; however, it has a higher half-life than CRP. Therefore its levels decline slower during an inflammation period, and can be influenced by various factors such as lifestyle factors and various metabolic abnormalities [92]. These factors, overlapping with the low specificity of inflammatory markers in IBD, make ESR a less reliable predictor of prognosis; however, it can yield good results in combination with CRP [93]. In addition, platelet count can be used as a serological marker in combination with ESR and CRP, as showcased by previous studies, highlighting increased reactivity in the presence of IBD [90,91,92,93]. However, it is not considered an independent predictor of inflammatory activity and cannot be used to discriminate CD from UC [90,92].

Recent studies focused on the predictive role of proinflammatory cytokines, which contribute to the inflammatory status of the intestine. Marafini et al. hypothesized that cytokines are good predictors of the response to TNF-alpha therapy [94]. Billiet et al. associated lower IL-8 serum levels with the primary infliximab therapy response. Moreover, a study found that low serum levels of IL-1B are associated with an infliximab response [95]. Even more so, Feng et al. observed that decreased IL-9 and IL-14 serum levels when compared with baseline levels can be predictors for mucosal healing [96]. In another study, the predictive value of pro-inflammatory cytokines in UC patients treated with vedolizumab was evaluated. Significant IL-6 and IL-8 decreases were associated with clinical remission, with a sensitivity and specificity of over 80% to predict clinical remission and mucosal healing [97].

### 3.4. Fecal Markers

Fecal markers are a series of substances secreted by the inflamed mucosa of the intestines and are considered a non-invasive way of measuring intestinal inflammation [98]. Despite fecal lactoferrin being the earliest fecal marker studied, back in 1960 [99], the discovery of fecal calprotectin paved the way for discovering further markers [100]. To this date, fecal calprotectin remains one of the most reliable markers in the diagnosis as well as monitoring of the progress of the disease. Neutrophil-derived proteins, PMN-elastase, lysosome, and myeloperoxidase are also elevated in the fecal matter of IBD patients [101,102,103,104]. One of the major advantages of fecal markers stands in their feasibility to be obtained, since fecal samples are accessible in any clinical setting. Another advantage stands in their reliability, since fecal matter comes into direct contact with the inflammation situs; therefore, the concentration of the specific markers can be used to gauge the severity of the IBD [105].

Calprotectin is a cytoplasmic protein, commonly found in neutrophils, consisting of more than 50% of its cytosolic components, as well as almost 5% of the total protein [106]. Its active conformation is a heterocomplex of S100A8 and S100A9. As it is a calcium and zinc binding metalloprotein, its primary roles are antibacterial, antifungal, inhibition of metalloproteinases, and apoptosis induction in damaged cells [107]. Recent studies have shown that it can bind transitional metals, steering them away from the pathogens to limit their growth, in a process called “nutritional immunity” [108,109]. Since calprotectin is released by neutrophils, its levels will increase in an inflammatory setting; therefore, it can be used to differentiate IBD from IBS [110].

Even though most of the laboratories use a normal value set at 50 μ/g, recent studies have shown the lack of specificity of the calprotectin, as it can increase in other gastrointestinal pathologies such as diverticulitis or bowel malignancy. Therefore, a clear clinical approach is required to adequately use calprotectin [111]. Lin et al. supports this cut-off value, which can be used for screening, with a sensitivity of over 90% and a specificity of 60% [112]. Menees et al. suggested that patients with clinical values under 40 μ/g with IBS symptoms are most unlikely to present IBD (with under 1% chance) [113]. However, several studies have noted that an increase in the cut-off value of calprotectin will increase the specificity, deeming it more appropriate for monitoring disease activity [114,115,116,117].

Fecal calprotectin presents a significant correlation with endoscopic activity in IBD, as highlighted in a meta-analysis by Rokkas et al. [118]. Endoscopic activity was correlated with fecal calprotectin, showcasing a sensitivity of 85% and a specificity of over 70% in diagnosing active IBD. When accounting for relapse prediction in IBD, CD, or UC, there is a variability in both sensitivity and specificity in the current literature. In a meta-analysis performed by Mao et al., combined over six studies, the pooled sensitivity and specificity were over 70%, with cut-off values varying from 100 to 340 μ/g [119]. In addition, fecal calprotectin can be used as a fecal marker in predicting postoperative recurrence, with cut-off values ranging from 50 to 200 μ/g. Liu et al. reported a pooled sensitivity and specificity of nearly 70%, with sensitivity values decreasing with the increase in the cut-off value [120]. Although fecal calprotectin presents a high degree of specificity and can be correlated with mucosal healing, which is pivotal in adequately monitoring IBD, it can still present some limits relating to adequate diagnosis, individual biological variations, as well as age. However, it remains one of the most accurate means of prognosis and follow-up as patients can be monitored through home-based testing with great turnaround times [121].

Although lactoferrin is present in most exocrine secretions, being an iron-binding glycoprotein and a component of polymorphonuclear neutrophiles, it can exhibit an increase in the fecal matter of the patients with IBD during the inflammatory processes. It can be detected using ELISA techniques due to its stability in fecal matter [122]. Primary studies have shown that fecal lactoferrin (FL) can be used in measuring the activity of the IBD as well as distinguishing an IBD from a non-inflammatory bowel disease [123]. Langhorst et al. surmised that FL can be a moderate predictor of sustained clinical remission and mucosal healing, with an area under the curve of 0.734 (CI 0.654–0.813, *p* < 0.0001) [124]. There is, however, a paucity of studies regarding the usage of FL in predicting relapse as well as postoperative recurrence in IBD. Yamamoto et al. highlighted a cut-off value for FL in relapse prediction with no statistically significant results [125]. Other two studies showed differences regarding relapse; however, the cut-off value could not be accurately predicted [126,127]. Even though a meta-analysis performed by Wang et al. suggests a good discriminative power between IBD and non-IBD patients, the role of fecal lactoferrin in the current clinical setting needs to be re-evaluated. Current studies suggest a combined approach, to increase both the sensibility and the specificity of the diagnosis as well as the prognostic role [128].

Myeloperoxidase, an enzyme also released during acute inflammation, can play an important role in the prognosis of IBD. A recent study published by Swaminathan et al., in which 172 participants were evaluated by providing biological samples as well as undergoing an ileocolonoscopy, showcased that fecal myeloperoxidase was effective in predicting moderate to severely active CD (AUC 0.86) and UC (AUC 0.92), thus being an adequate biomarker of endoscopic activity. Thus, it might be used to evaluate the treatment outcome [129].

S100A12 or calgranulin C is a cytoplasmic protein with a proinflammatory function and chemotactic activity, which can activate the nuclear factor-kB. Its expression is enhanced by various proinflammatory factors, which suggests that it contributes to the process of intestinal inflammation [130]. S100A12 is overexpressed in inflammatory conditions [131] and it also has a half-life of over 7 days in the bowel, which makes it an ideal biomarker. Initial studies in the early 2000s suggested an overexpression of S100A12 in inflammatory bowel disease, able to distinguish between IBD and IBS [132,133]. Although there is a paucity of studies regarding the discrimination between UC and CD, recent studies focused on the prognostic role in IBD. Däbritz et al. showed that a cut-off value of 0.43 μg/g was able to predict relapse 8–12 weeks earlier with sensitivity and specificity being 70 and 83% respectively [134]. Wright et al. examined S100A12 in pediatric patients who underwent surgery, to evaluate postoperative recurrence risk. Despite obtaining high sensitivity (over 90%), no significant differences were observed in patients with and without postoperative recurrence [135].

Lipocalin is a gelatinase-associated marker, of 25 kDa, with high expression in neutrophils and adipocytes found in the gastrointestinal, urogenital, and respiratory tract [136]. It has bacteriostatic effects by sequestering iron, and it can act as a growth factor by stabilizing the proteolytic enzyme MMP-9 [137,138]. It is strongly expressed in the intestinal cell layer during inflammation. Chassaing et al. emphasized its role in intestinal inflammation in an experimental model on mic [139]; moreover, Yadav et al. highlighted its sensitivity to discriminate intestinal inflammation in multiple sclerosis. Although the studies of fecal lipocalin are relatively limited, the characteristics of lipocalin make him an ideal candidate marker for evaluating inflammatory bowel diseases [140].

Another novelty biomarker is neopterin, an intermediate metabolite of biopterin, which is released from activated macrophages. A study by Nancey et al. in 2013 highlighted that the diagnostic accuracy of neopterin was comparable with calprotectin, evidencing moderate sensitivity. In addition, fecal neopterin correlates well with the severity of the lesions [141]. Further studies are required to cement neopterin’s role in the prognostic of inflammatory bowel diseases.

Other fecal markers, though not widely studied, may prove to be of significant importance. Other fecal metalloproteinases (MMPs) were highlighted by several studies, in which the levels were significantly higher than in the control population [142,143,144]. Their role was adequately described in a review by Maronek et al. in 2021 [145]. M2-Piruvate Kinase has been shown to increase in IBD, in some studies managing to differentiate IBD from IBS [146,147]. Other markers have been taken into consideration; however, they present low sensitivity or specificity, or they require invasive procedures to be collected, a notable example of that being fecal nitric oxide.

### 3.5. Other Non-Invasive Methods of Evaluating IBD

Although in current clinical practice clinical variables, endoscopic scores, and accessible biomarkers remain the pinnacle of diagnosis in IBD, there are still certain limitations regarding individual treatment, cost effectiveness, and precision. Attempting to overcome these limitations, through collaborative efforts, an increased number of multiomic projects have been developed.

The development of IBD is strongly linked with the environmental as well as microbiota interactions of the digestive system. This link is the current array of metabolomics in which the pathogenic mechanisms of IBD can be researched. De Peter et al. highlights the role of metabolomics as well as its potential impact and implications [148]. Metabolic profiles are being performed and analyzed to highlight differences between the healthy and affected patients. Williams et al. highlighted differences between CD and UC profiles through H magnetic resonance spectroscopy [149]. The major drawback of metabolomics in its current state is that there is no strong evidence in predicting the accuracy of the models; therefore, more research is needed for this to be put in clinical practice [150].

To further develop the field of multiomics, several cohorts are developed to further integrate the multiomic data in order to minimize the heterogeneity of the IBD. Strong national data banks are created which contain information regarding genetics, inflammatory markers, metabolomics, and microbiomes in which thousands of patients with IBD are enrolled. Agrawal et al. highlights the importance of multiomics as well as discussing the benefits of network analysis from which data of clinical relevance can be derived [151]. Other such network analyses can open the pathway for discovering new protein interactions, microbiome–metabolomic networks, as well as gene networks [152,153,154].

Recent developments in artificial intelligence have determined an increased interest in implementing AI data mapping as well as data gathering and evaluation in the field of medicine. Recent studies regarding the implementation of machine learning for developing prediction models in IBD have been developed. A review of 13 studies by Nguyen et al. in 2022 highlighted that machine-learning-based prediction models perform better than traditional statistical models in risk prediction in IBD; however, an expansion of the studies is required in order to minimize the risk of bias and increase its clinical significance [155].

Furthermore, more elements of non-invasive methods of diagnosis and prognosis are being developed, one of them being Raman spectroscopy, an inelastic light-scattering phenomenon in which the illumination of a molecule by a mono-chromatic laser will create an exchange of vibrational energy. This provides a vibrational spectrum that contains information relative to the chemical bonds and symmetry of a specific molecule [156]. The introduction of Raman techniques as well as different technological modifications of the Raman apparatus have managed to create a significant development in defining tissues and the underlying processes as well as a variation in methods within the technique itself. Saletnik et al. highlighted the most noteworthy methods of Raman spectroscopy [157]. Surface-Enhanced Raman Spectroscopy (SERS) is based on the amplification of scattered Raman particles absorbed on the metal surface, characterized by an increase in the cross-section of the Raman scattering of the analyte. Colloid metals or rough electrodes are the most used active surfaces in SERS. Its role is primarily in biological research and nanotechnology, a major limitation currently being the fluorescence of some biological components. Another commonly used method is Confocal Raman Spectroscopy (CRM) which can also provide information regarding the internal structure of the material, able to provide images in three different axes (x, y, and z) at a micrometer level. Ultraviolet Resonance Raman Spectroscopy (UVRS) can selectively enhance Raman signal from chromophores, as the majority of molecules have an absorption band in the ultraviolet region, thus being an excellent tool in finding interactions between membrane proteins and lipid layers. 

Despite the studies being scarce in the field of inflammatory bowel diseases, an article published by Pence et al. in 2017 highlighted that, in combination with standard endoscopic evaluation, Raman spectroscopy may have the potential of becoming a diagnostic adjunct in IBD [158]. Table 4 highlights the studies in which Raman spectroscopy is used in regard to IBD detection and prognosis [158,159,160,161,162,163,164,165,166,167,168,169,170]. Bi et al. developed spectral markers for the discrimination of UC and CD demonstrating a higher content of lipid and lower amount of phenylalanine in the samples of patients with UC [159]. In addition, Bielecki et al. was able to discriminate patients with UC from patients with CD and healthy patients, in a cohort of 38 patients using Raman spectroscopic histopathology with a classification rate of 98% [160]. Wood et al. obtained similar results in the discrimination of various colorectal pathologies, with an accuracy of over 95% for 10 s acquisitions. These studies were supported by three more articles [161,162,163] which presented high levels of sensitivity (over 80%) in distinguishing IBD from healthy patients. The primary limitation of the studies, however, is that they had no degree of comparison with other prognostic methods in IBD. In 2020, Kirchberger et al. highlighted a high degree of sensitivity and specificity in comparison with endoscopic scores (Se = 78%, Sp = 93%) [167]. Recent studies focused on the utilization of SERS in discriminating both CD as well as UC, yielding similar results [168,169,170]. 

Smith et al. highlighted, in a cohort study of 23 patients, that Raman spectroscopy can highlight biochemical changes following the treatment of IBD and can accurately differentiate mucosal healing from inflammation, thus being an adequate diagnostic tool [171]. Despite the main limit being that the previously mentioned studies resorted to a biopsy obtained via a colonoscopy, in a study published by Acri et al., proteic extract from the fecal matter of fifteen pediatric subjects with CD, nine subjects with UC, and nineteen subject in which IBD was ruled out were analyzed, yielding great results in differentiating the diseases, thus highlighting the potential for a non-invasive diagnostic tool [172]. Further research regarding the usage of Raman spectroscopy in serum biomarkers in IBD is required.

### 3.6. Imaging Methods in Inflammatory Bowel Diseases

The current practices of diagnosing IBD are a combination of radiological, endoscopic, histological, and biochemical procedures, since none of these can be considered a reliable standalone procedure in the diagnosis as well as the prognosis of it [173]. Since IBD, especially CD, can affect the digestive tract on many levels, it can prove very difficult to assess and adequately evaluate the lesions. Therefore, the previous standard of evaluation in IBD was considered barium fluoroscopy, which could highlight abnormalities in the mucosal pattern. However, the primary limitation was the lack of transmural evaluation as well as other extraluminal complications of CD.

Computer-Tomography Enterography gained traction in the past 15 years, having the particular advantages of producing high-resolution tridimensional images and highlighting any problems at the bowel wall as well as the presence of extraintestinal complications, presence of fistulas, or lymph node involvement. Paulsen et al. reported good results over a cohort of 700 cases in discriminating small bowel CD from other intestinal diseases [174]. One of the primary drawbacks of CT enterography, however, remains the risk of exposure to ionizing radiation which can be extremely relevant in pediatric or younger patients [175].

For all the considerations above, MRI has gained significant traction in the last decade, proving a higher degree of sensitivity in detecting small bowel activity of CD, as well as limiting the radiation exposure [176]. In addition, MRI has a higher sensitivity in detecting transmural fibrosis as well as in highlighting early clinical responses [177,178].

Despite the fact that the role of imaging has been widely discussed in previous studies as well as reviews and guidelines [179], there are still some remaining issues regarding the sensitivity of the imaging procedures, as well as the aforementioned risks and costs. Haas et al. highlighted the importance of ultrasound in the diagnosis of IBD as well as a step-up approach in using radiology, in order to maintain cost-effectiveness as well as minimizing the risk of radiation [180].

## 4. Conclusions

There is a constant need to improve the methods of detection as well as prognosis of inflammatory bowel disease, with a tendency towards being minimal to non-invasive. To this date, CRP and FC remain the most reliable markers of intestinal inflammation as well as prognosis in IBD, due to their accessibility, cost effectiveness, as well as specificity. Despite their accuracy, there are limitations for the markers, in terms of reliability and individual cases. A more adequate solution, achieved by obtaining a combination of biomarkers to increase the accuracy and prediction possibility as well as maintaining costs, is required. An early diagnosis as well as an early prognosis will lead to a better quality of life as well as minimizing the costs and potentially inducing an individual into clinical remission. The highlighted markers can serve as a potential steppingstone into developing new prognostic non-invasive methods. In addition, the current methods of diagnosis and prognosis can be, furthermore, combined to achieve the most effective, feasible, and cost-effective method of diagnosis with the highest degree of reliability and accuracy. In addition, the currently available information can pave the way for adequate research, narrowing or selecting the markers or methods which yield the best results.

With an ever-changing climate as well as rapid development in most countries, inflammatory bowel diseases maintain their complexity in terms of geography, gender, and microbiota. Therefore, all cases must be treated individually, which is the primary limitation of the non-invasive biomarkers. Artificial intelligence can provide further insight into an adequate selection of the combination of markers and can map certain points which may be overlooked by current researchers. The recently developed field of multiomics may allow the development of individualized, sensitive, as well as specific biomarkers for a much more adequate monitoring of IBD. Further research must take into consideration the turning points of IBD: need for surgery, mucosal healing, hospitalization duration, disease flares, and costs. Multicentric studies are required to determine the reliability of both serum and fecal biomarkers as well as to develop adequate analyses in the field of multiomics to develop further methods of diagnosis and prognosis while pertaining to the quality of life of the patient.

## Figures and Tables

**Table 1 ijms-25-02077-t001:** Crohn Disease Activity Index.

Variable	Coefficient
Number of liquid stools each day for seven days	×2
Abdominal pain each day for seven days (0–3 grade of severity)	×5
General well-being each day for seven days (0–4 grading)	×7
Complications	×20
Use of opioids for diarrhea	×30
Abdominal mass (0—none, 2—questionable, and 5—definitive)	×10
Deviation in hematocrit	×6
Deviation in weight	×1

**Table 2 ijms-25-02077-t002:** Truelove–Witts Score.

Parameters	Mild	Moderate	Severe
Bloody stools per day	<4	4 to 6	>6
Pulse	<90 bpm	<90 bpm	>90 bpm
Temperature	<37.5 °C	<37.8 °C	>37.8 °C
Hemoglobin	>11.5 g/dL	>10.5 g/dL	<10.5 g/dL
ESR	<20 mm/h	<30 mm/h	>30 mm/h
CRP	Normal	<30 mg/dL	>30 mg/dL

**Table 3 ijms-25-02077-t003:** Crohn Disease Endoscopic Index of Severity.

Variable	0	1	2	3
Size of ulcers	None	<0.5 cm	0.5–2 cm	>2 cm
Percentage of ulcerated surface	None	Less than 10%	10–30%	More than 30%
Percentage of affected surface	None	Less than 50%	50–75%	More than 75%
Presence of stenosis	None	Single, passable	Multiple, passable	Impassable

**Table 4 ijms-25-02077-t004:** Recent studies involving Raman spectroscopy for differentiation and detection of IBD.

Authors	Year	Title	Outcomes
Bi et al. [159]	2011	Development of spectral markers for the discrimination of ulcerative colitis and Crohn’s disease using Raman spectroscopy	High differentiation of nucleic acid, phenylalanine and lipid spectra between the cohorts
Bielecki et al. [160]	2012	Classification of inflammatory bowel diseases by means of Raman spectroscopic imaging of epithelium cells	High degree of differentiation between IBD and healthy patients (Se = 99.07, Sp = 98.81)
Beleites et al. [161]	2013	Raman spectroscopy and imaging: promising optical diagnostic tools in pediatrics	Despite small sample size, high degree of differentiation between IBD patients
Pence et al. [164]	2014	Endoscopy-coupled Raman spectroscopy for in vivo discrimination of inflammatory bowel disease	Moderately high accuracy of discrimination of IBD from healthy patients (Acc = 79.7%)
Wood et al. [162]	2014	Evaluation of a confocal Raman probe for pathological diagnosis during colonoscopy	High degree of separation between colo-rectal pathologies (Acc = 95%)
Veenstra et al. [163]	2015	Raman spectroscopy in the diagnosis of ulcerative colitis	Good specificity and sensibility in discrimination of patients with UC (Se = 82%, Sp = 89%)
Pence et al. [158]	2017	A. Clinical characterization of in vivo inflammatory bowel disease with Raman spectroscopy	High sensitivity and moderate specificity in discriminating IBD (Se = 85%, Sp = 58%)
Ding et al. [165]	2020	In vivo analysis of mucosal lipids reveals histological disease activity in ulcerative colitis using endoscope-coupled Raman spectroscopy	Accurate endoscopic discrimination of UC from healthy controls (Se = 83.5%)
Morasso et al. [166]	2020	Raman Analysis Reveals Biochemical Differences in Plasma of Crohn’s Disease Patients	Good classification of CD from healthy controls (Se = 80%, Sp = 85.7%)
Kirchberger et al. [167]	2020	Towards an Interpretable Classifier for Characterization of Endoscopic Mayo Scores in Ulcerative Colitis Using Raman Spectroscopy	Good sensitivity and specificity in comparison with Endoscopic Mayo Scores (Se = 78%, Sp = 93%)
Tefas et al. [168]	2021	Surface-enhanced Raman scattering for the diagnosis of ulcerative colitis: will it change the rules of the game?	High sensitivity, specifcity and AUC in discriminating UC from healthy patients (Se = 94%, Sp = 92%, AUC = 0.96)
Li et al. [169]	2021	Non-invasive diagnosis of Crohn’s disease based on SERS combined with PCA-SVM	High sensitivity and specificity in discriminating CD from healthy patients (Se = 86%, Sp = 87.5%)
Buchan et al. [170]	2023	Raman spectroscopic molecular fingerprinting of biomarkers for inflammatory bowel disease	Raman spectroscopy is a rapid, non-invasive technique for multiplex profiling, yielding an efficient combination of specific potential IBD indicators

## Data Availability

No new data were created or analyzed in this study. Data sharing is not applicable to this article.

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
