# Peer review of "Recent Trends in Non-Invasive Methods of Diagnosis and Evaluation of Inflammatory Bowel Disease: A Short Review"

_ijms, 2024, doi:10.3390/ijms25042077_

Round 1
Reviewer 1 Report
Comments and Suggestions for Authors
Manuscript needs revision.

Minor editing is required.
Author Response
Greetings and thank you for your in-depth answer as well for the valuable feedback that we took into consideration. Below you have the point-by-point suggestions.
- We indeed agreed that the abstract felt a bit underwhelming, therefore we rewrote it and tried to restructure it to showcase its actual purpose of finding trends or possibilities of early, quick, cost-effective and precise methods of diagnosis. We thank you for suggesting that.
- We’ve included the suggested informations in the introduction; although the possible implications and the effect of individuals are discussed in the discussion/conclusions section, we decided to pinpoint some aspects in the introduction as well.
- We’ve had some issues uploading the tabulations. We provided the tabulations accordingly, as a supplementary material since we had issues with the formatting.
- We’ve added a section regarding the platelet count as well, despite not being the most specific marker, we felt that it should not be disregarded.
- There’s already a section regarding miRNA, and we felt that it has already been reviewed by other authors. If required, we can expand on it, as per your suggestion.
- We’ve attempted to take a shift from the conventional methods of diagnosis; however, we agree that imaging plays one of the most important roles in the NON-INVASIVE diagnosis, therefore we added a section for it. If required we can further expand it.
- We have expanded the paragraph regarding the prospects in IBD, as well as highlighting the current databases and IBD trials which can further develop the methods of prognosis and diagnosis.
- This issue is like point 3. We tried adding it accordingly, however due to the current format, we were not able to input the table in our article without modifying its entire structure. We will upload the tables separately.
Minor comments:
- We rewrote the entire paragraph, as we can agree it did not sound quite right (from an English point of view as well as a medical point of view).
- We have abbreviated the most used syntaxes (IBD, CD, UC) and we made a glossary of abbreviations at the end of the article.
Reviewer 2 Report
Comments and Suggestions for Authors
Authors can find comments below:
1. Is this a systematic study? The structure of the paper is confusing. Please state clearly in the abstract or somewhere before the main body of the manuscript.
2. The author utilizes the Crohn’s Disease Activity Index, where the index's variability, reliability, and limitation must be briefly explained.
3. Please tabulate the primary information in the table form for Serological Markers, Blood inflammatory markers, and fecal marker
4. Please expand on the statement "Raman spectroscopy is used in regard for IBD detection and prognosis." You may consider references from 157-169.
5. How Raman spectroscopy is helpful in diagnosis needs more description, which is understated in the current form. For example, Raman spectroscopy can be different, depending on the sample and what kind of biomolecules/forms are investigated, such as UV-Raman Spectroscopy for aromatic compounds (like natural polyphenols, lignin, etc.); confocal Raman Spectroscopy for mapping; Time-gated Raman spectroscopy for biological markers or biotags.
Author Response
Greetings and thank you for your in-depth suggestions as well as some QoL improvements. Below we have managed to answer all of your suggestions. Suggestions and reworks are marked red in the text.
- Is this a systematic study? The structure of the paper is confusing. Please state clearly in the abstract or somewhere before the main body of the manuscript
Although our main intention was to create a brief review, we couldn’t pluck out a certain number of articles as it would’ve created a selection bias (cherry-picking). We decided to switch the structure to a systematic study, therefore as per your suggestion we stated it clearly at the start of the manuscript.
- The author utilizes the Crohn’s Disease Activity Index, where the index's variability, reliability, and limitation must be briefly explained.
We’ve already highlighted that, since CDAI is a non-invasive measurement of CD its reliability is only available when combined with other methods of prognosis. There are modified scores which can be used in clinical trials in order to obtain a rapid response and be a subject for self-evaluation. This part, as well as the limitations are highlighted in the study (With references).
- Please tabulate the primary information in the table form for Serological Markers, Blood inflammatory markers, and fecal marker
We’ve had some issues uploading the tabulations. We provided the tabulations accordingly, as a supplementary material since we had issues with the formatting (see Table 4, which we deleted entirely). We provided the tabulations for the requested information as per your suggestions and we thank you for highlighting that.
- Please expand on the statement "Raman spectroscopy is used in regard for IBD detection and prognosis." You may consider references from 157-169.
- How Raman spectroscopy is helpful in diagnosis needs more description, which is understated in the current form. For example, Raman spectroscopy can be different, depending on the sample and what kind of biomolecules/forms are investigated, such as UV-Raman Spectroscopy for aromatic compounds (like natural polyphenols, lignin, etc.); confocal Raman Spectroscopy for mapping; Time-gated Raman spectroscopy for biological markers or biotags.
As a part of our research will be focus on the Raman Spectroscopy, we decided that some in-depth explanations (as per your surggestions) are required. Thus we rewrote the paragraphs to be more suitable for an open discussion as well as highlighting the progress made by our peers.